# Effect of Temperature on the Development of Stages of Spermatogenesis and the Functionality of Sertoli Cells In Vitro

**DOI:** 10.3390/ijms25042160

**Published:** 2024-02-10

**Authors:** Areej Jorban, Eitan Lunenfeld, Mahmoud Huleihel

**Affiliations:** 1The Shraga Segal Department of Microbiology, Immunology, and Genetics, Faculty of Health Sciences, Ben-Gurion University of the Negev, Beer Sheva 8410501, Israel; areejj@post.bgu.ac.il; 2The Center of Advanced Research and Education in Reproduction (CARER), Faculty of Health Sciences, Ben-Gurion University of the Negev, Beer Sheva 8410501, Israel; 3Adelson School of Medicine, Ariel University, Ariel 4076414, Israel; eitanlun@ariel.ac.il

**Keywords:** temperature, organoids, spermatogenesis, spermatogonial stem cells, in vitro differentiation of spermatogonial cells, Sertoli cells

## Abstract

Spermatogenesis is the process of proliferation and differentiation of spermatogonial cells to meiotic and post-meiotic stages and sperm generation. Normal spermatogenesis occurs in vivo at 34 °C to 35 °C, and high temperatures are known to cause male infertility. The aim of the present study was to examine the effect of temperature (35 °C compared to 37 °C) on the viability/apoptosis of developed cells, on the development of different stages of spermatogenesis in 3D in vitro culture conditions, and the functionality of Sertoli cells under these conditions. We used isolated cells from seminiferous tubules of sexually immature mice. The cells were cultured in methylcellulose (as a three-dimensional (3D) in vitro culture system) and incubated in a CO_2_ incubator at 35 °C or 37 °C. After two to six weeks, the developed cells and organoids were collected and examined for cell viability and apoptosis markers. The development of different stages of spermatogenesis was evaluated by immunofluorescence staining or qPCR analysis using specific antibodies or primers, respectively, for cells at each stage. Factors that indicate the functionality of Sertoli cells were assessed by qPCR analysis. The developed organoids were examined by a confocal microscope. Our results show that the percentages and/or the expression levels of the developed pre-meiotic, meiotic, and post-meiotic cells were significantly higher at 35 °C compared to those at 37 °C, including the expression levels of the androgen receptor, the FSH receptor, transferrin, the androgen-binding protein (ABP), and the glial-derived nerve growth factor (GDNF) which were similarly significantly higher at 35 °C than at 37 °C. The percentages of apoptotic cells (according to acridine orange staining) and the expression levels of BAX, FAS, and CASPAS 3 were significantly higher in cultures incubated at 37 °C compared to those incubated at 35 °C. These findings support the in vivo results regarding the negative effect of high temperatures on the process of spermatogenesis and suggest a possible effect of high temperatures on the viability/apoptosis of spermatogenic cells. In addition, increasing the temperature in vitro also impaired the functionality of Sertoli cells. These findings may deepen our understanding of the mechanisms behind optimal conditions for normal spermatogenesis in vivo and in vitro.

## 1. Introduction

Spermatogenesis is a process that occurs in the seminiferous tubules, where spermatogonial cells proliferate and differentiate into meiotic and post-meiotic stages that continue the process of spermiogenesis to generate mature sperm [1]. This process is controlled by germ cell–Sertoli cell interactions and other somatic cells such as peritubular cells and Leydig cells that together provide the optimal conditions for the completion of spermatogenesis [2,3,4]. Endocrine hormones such as gonadotropins and testosterone are additionally involved in this process [5]. Luteinizing hormone (LH) acts on Leydig cells and induces the production of testosterone that affects the functionality of Sertoli and peritubular cells [6,7]. Follicular stimulating hormone (FSH) affects different functional markers of Sertoli cells such as the androgen binding protein (ABP), the androgen receptor (AR), and inhibin [8], as well as testicular para/autocrine factors such as colony-stimulating factor-1 (CSF-1), the glial-cell-derived nerve growth factor (GDNF) and the leukemia inhibitory factor (LIF) [9,10].

The process of spermatogenesis in vivo is a temperature-dependent process. Spermatogenesis occurs optimally at temperatures slightly lower (2–3 degrees) than that of the body [11,12,13]. Elevated temperatures (37 °C to 39 °C) have a substantial impact on spermatogenesis and the generated spermatozoa, potentially resulting in loss of male fertility [14,15]. These findings are clearly supported by several studies demonstrating that elevated temperature in the scrotum/testicles of fertile men reduces both sperm output and quality [15]. Also, studies have consistently revealed that heightened testicular and epididymal temperature can severely damage sperm morphology in humans [16], and exposure of the testes to body temperature due to local testicular hyperthermia, cryptorchidism, and varicocele increases germ cell death [17]. Various in vivo studies that examined the heat sensitivity of spermatogenesis in cryptorchid testes showed that high temperatures cause spermatogenesis imperfection, including induction of germ cell death through apoptosis [14,18].

Furthermore, recent studies showed that somatic cell maturation and spermatogonial potential differentiation of rat testicular organoids were higher when incubated at 34 °C than when incubated at 37 °C [19]. Additional studies using in vitro culture models of spermatogonial stem cells (SSC) have demonstrated that high temperatures (37 °C or 43 °C) significantly inhibited SSC differentiation through P53, ribosomes and carbon metabolism signaling pathways, changing their transcriptome [20]. Furthermore, previous studies have explored the possible impact of high temperatures on miRNA expression that may impair the spermatogenesis process [21,22].

Research involving bulls and deer has also substantiated the correlation between increased testicular and epididymal temperatures and reduced sperm parameters, including motility, acrosomal integrity, and morphology [15,23]. In a study utilizing ex vivo cultures of mouse testis, spermatogenesis failed at multiple steps at a temperature of 38 °C/37 °C, including damage to the first meiotic prophase, increased DNA double-strand breaks (DSBs), and compromised DSB repair [24]. Moreover, high temperatures (37 °C) have been shown to directly affect Sertoli cell functions and cause Sertoli cells to produce less ABP and display morphological damage [25].

One of the methods for male fertility preservation, especially in prepubertal boys, is the establishment of in vitro culture systems for the generation of fertile sperm [26], and this technology is used to mimic in vivo conditions. Our group established a novel three-dimensional (3D) in vitro culture system using a methylcellulose culture system (MCS) and a soft agar culture system (SACS) and demonstrated the induction of spermatogonial cells in vitro for different stages of spermatogenesis in mice, monkeys, and humans [26,27]. These systems were cultured in a CO_2_ incubator at 37 °C and mostly induced the development of haploid cells/round spermatids [28]; in some cases, they induced the development of sperm-like cells [29]. On the other hand, our 3D in vitro culture system conditions were not optimal to develop consistently complete spermatogenesis. According to our published studies [26,28,30] and to improve our in vitro culture system to generate sperm, we examined the possibility of decreasing the in vitro incubation temperature to a similar temperature in the testis (in vivo).

In the present study, we evaluated the possible effect of temperature on the viability of cultured cells, the development of different stages of spermatogenesis, and the functionality of Sertoli cells in MCS. We compared cultures with testicular temperature (in vivo, 35 °C), body temperature (37 °C) and incubation period of development of spermatogenesis in mice (until 6 weeks—time of development of complete spermatogenesis in vivo).

## 2. Results

### 2.1. Effect of Temperature on Apoptosis of the Growing Cells In Vitro

Isolated cells from seminiferous tubules of seven-day-old mice were cultured in vitro using MCS and in the presence of StemPro and knock-out serum replacement (KSR) at different temperatures (35 °C and 37 °C) and for different lengths of time (two, four and six weeks). Our results showed that spheroids/organoids that developed at different temperatures did not show differences in their size and shape (Figure 1A).

Developed spheroids/organoids were collected four weeks after culture and stained with propidium iodide and acridine orange before (Figure 1(B.1,C.1)), and enzymatic treatment was performed to separate the cells and break up the colonies (Figure 1(B.2,C.2)).

Our results show that the percentage of apoptotic cells is significantly higher when cultured at a temperature of 37 °C compared to the percentage of those cultured at a temperature of 35 °C (*p* = 0.0001), and the percentage of viable cells is significantly higher at 35 °C compared to the percentage of those cultured at 37 °C (*p* = 0.002) (Figure 1D). In addition, the expression levels of the apoptotic markers FAS, BAX and CASPAS3 are significantly higher at 37 °C than at 35 °C at each examined time point (Figure 1(E.1–E.3), respectively).

### 2.2. In Vitro Development of Organoid-like Structure Shows a Cellular Composition Similar to the Seminiferous Tubules

The developed organoid-like structure was collected after culturing at 37 °C and 35 °C and stained by immunofluorescence (IF) staining using specific markers for Sertoli cells (vimentin) and for seminiferous tubule basement membrane (BM) (collagen IV) as well as specific markers of meiotic and post-meiotic stages (CREM and ACROSIN, respectively) to be evaluated by confocal microscopy. Our results show that the organoid-like structures composed of Sertoli cells and seminiferous tubule BM developed at both temperatures (37 °C and 35 °C) (Figure 2(A1,A2)), in a similar manner to the in vivo structure of the seminiferous tubule (Figure 2E,F). Furthermore, we also showed the presence of Sertoli cells and germ cells at the meiotic stage at both temperatures (37 °C and 35 °C) and that the meiotic cells are present in between the Sertoli cells at both temperatures (37 °C and 35 °C) (Figure 2(B1,B2)), in a similar cellular structure to that of the seminiferous tubules in vivo (Figure 2F). In addition, the organoid-like structures were positively stained for ACROSIN alone and for both CREM and ACROSIN at both culture temperatures (37 °C and 35 °C) (Figure 2(C1) and (C2), respectively), shown in Figure 2(D1) and (D2), respectively. Furthermore, we were able to identify organoid-like structures with positive staining for peritubular cells.

Alpha-smooth muscle actin; ASMA outside the organoid, Sertoli cells with positive staining for SOX9 inside the organoid and a lumen inside the organoid-like structure are presented (Figure 2G). In addition, we demonstrated the presence of Leydig cells (cells that positively stained for 3βHSD; a specific marker for Leydig cells) outside the organoid-like structure (Figure 2H).

### 2.3. Effect of Temperature on the Development of Different Stages of Spermatogenesis In Vitro

Developed cells in MCS at 37 °C and 35 °C were collected after two to six weeks of culture, after which they were fixed and stained with antibodies specific for cell markers of the different stages of spermatogenesis. For pre-meiotic stages, we used anti-VASA and anti-GFR-α antibodies to identify cells at the pre-meiotic stage. For the meiotic stage, we used anti-Boule antibodies to identify cells at the meiotic stage, and for the meiotic/post-meiotic stage, we used anti-ACROSIN antibodies to identify cells at the meiotic/post-meiotic stage. Our results showed that cells before culture (BC) contained only cells of the premeiotic stages (VASA and GFR-α) (Figure 3). However, in cultures that grew in MCS for five weeks at 37 °C and 35 °C, we found cells of the pre-meiotic (VASA and GFR-α-positive cells), meiotic (BOULE-positive cells) and post-meiotic stages (ACROSIN-positive cells) (Figure 3).

### 2.4. Effect of Temperature on the Development of the Pre-Meiotic VASA and GFR-Alpha Positive Cells In Vitro

Our results indicate that the in vitro culture of isolated seminiferous tubule cells in MCS for two to six weeks at different temperatures (35 °C and 37 °C) significantly increased the percentage of the stained cells (VASA/GFR-α) compared to the data obtained before culture (0) (Figure 4(A.1,B.1)) (the positive-stained cells for each examined marker were counted from at least a total of 100 cells present in the examined field. In each field/examined sample, we have a lot more positive- and negative-stained cells).

After two and six weeks of culture, the percentage of VASA- and GFR-α-positive-cells was higher at 35 °C than at 37 °C. However, after four weeks, there was no difference in the percentage of VASA cells, while there was a significant increase in GFR-α cells at 35 °C than at 37 °C (Figure 4(A.1,B.1)).

In addition, the expression levels of VASA were significantly higher at 35 °C than at 37 °C after two and six weeks of culture, and there was no difference after four weeks (Figure 4(A.2)) (in a similar manner to the percentage of the stained cells). However, the expression levels of GFR-α were significantly lower at 35 °C than at 37 °C after two weeks of culture, and there was no difference after four and six weeks of culture (Figure 4(B.2)) (this is different from the percentage of the stained cells).

### 2.5. Effect of Temperature on the Development of the Meiotic and Post-Meiotic Cells In Vitro

As expected, our results showed that BC, there were no meiotic/post-meiotic cells (BOULE and ACROSIN) present. However, their growth in the culture (two to six weeks) at 37 °C and 35 °C induced the percentages of meiotic cells (BOULE) and meiotic/post-meiotic cells (ACROSIN) in comparison to BC (Figure 5(A.1) and (B.1), respectively).

After two and four weeks of culture, the percentage of BOULE-positive cells was higher at 35 °C than at 37 °C (Figure 5(A.1)), and the percentage of ACROSIN-positive cells was higher after four weeks at 35 °C than at 37 °C (Figure 5(B.1)). However, after six weeks, the percentages of BOULE- and ACROSIN-positive cells were similar in cultures that grew at 35 °C and 37 °C (Figure 5(A.1,B.1)).

In addition, the expression levels of BOULE-positive cells were higher at 35 °C than at 37 °C after two and four weeks of culture (Figure 5(A.2)), and the expression levels of CREM were higher at 35 °C than at 37 °C only at four weeks (Figure 5C). Furthermore, the expression levels of meiotic/post-meiotic markers significantly decreased in cultures that grew for two weeks at 35 °C compared to those that grew at 37 °C, while the expression levels of ACROSIN and PROTAMIN were significantly higher in cultures that grew for four weeks at 35 °C compared to those that grew at 37 °C (Figure 5(B.2) and D, respectively). On the other hand, there was no significant effect of temperature on the expression levels and percentage of meiotic/post-meiotic cells that grew for six weeks in the cultures (Figure 5A–D).

### 2.6. Effect of Temperature on the Development of Haploid Cells In Vitro

Developed cells and organoids were collected four weeks after culture, under conditions of 37 °C and 35 °C, and were enzymatically disassociated. The cells were stained with Hoechst 33342 and analyzed by flow cytometry (Figure 6(A.3,A.4)). The presence of tetraploid, diploid, and haploid cells were examined in all samples. Our flow cytometry results show that isolated cells from seminiferous tubules of immature mice (used as a negative control for haploid cells) contained only 2N and 4N cells (Figure 6(A.1)). However, cells isolated from seminiferous tubules of mature mice (used as a positive control for haploid cells), and from the culture that incubated at 37 °C and 35 °C contained 1N, 2N and 4N cells (Figure 6(A.2–A.4)).

Quantification of the results from the culture showed that the percentage of 4N cells from cultures at both 37 °C and 35 °C decreased compared to those BC and that the percentages of 4N cells were similar in the cultures grown at 37 °C and 35 °C (Figure 6B). In addition, the percentage of 2N cells significantly increased in cultures grown at 37 °C and 35 °C compared to those grown BC (Figure 6B). Furthermore, the percentages of 2N cells were similar in cultures grown at 37 °C and 35 °C (Figure 6B). In addition, our results showed that the development of haploid cells in the culture was similar in cultures grown at 37 °C and 35 °C (Figure 6B). These results confirm our IF staining results related to the presence of meiotic and post-meiotic cells (haploid cells) in the cultures that grew at 37 °C and 35 °C.

To further study the development of round spermatids in our cultures, we examined the expression of round spermatid markers such as ACRV1 and SUN3. Our results show the expression of those markers in cultures that grew for two to six weeks at 37 °C and 35 °C while the higher expression levels were found in four-week-old cultures at 35 °C (Figure 6C,D).

### 2.7. Effect of Temperature on the Development and Functionality of Sertoli Cells In Vitro

After demonstrating the effect of temperature on the development of the various stages of spermatogenesis in vitro, our subsequent objective was to examine its potential influence on the functionality of Sertoli cells present in in vitro cultures. Therefore, we assessed the percentage of Sertoli cells present at each temperature and the RNA expression level of different markers related to Sertoli cell functionality, such as ABP, INHIBIN, GDNF and AR. Our results showed that although the percentage of Sertoli cells was higher at 37 °C than at 35 °C (Figure 7), there was a significant increase in the expression levels of ABP, INHIBIN, GDNF and AR in the cultures that grew at 35 °C compared to those that grew at 37 °C.

## 3. Discussion

Our results show that the incubation of isolated cells from seminiferous tubules of immature mice in vitro at a temperature close to in vivo testicular temperature (35 °C) increased the viability of the cells compared to the viability of those incubated at 37 °C. Furthermore, the percentages of apoptotic cells (according to AO staining) and the expression levels of BAX, FAS and CASPAS 3 in cultures incubated at 37 °C were higher than the levels of cultures cultured at 35 °C. These results are consistent with previous studies from other labs, which showed that high temperature caused germ cell death through apoptosis [14,18,31].

In addition, our study illustrated in vitro development of organoids with a cellular composition similar to seminiferous tubules cellular structures. The organoids were composed of meiotic/post-meiotic cells (CREM, ACROSIN) along with Sertoli cells (vimentin-positive cells) and seminiferous tubule BM, in which collagen type IV is one of the major components [32]. Furthermore, the branches of Sertoli cells enveloped the developing germ cells (the meiotic cells; CREM-positive stained cells) and collagen IV surrounded them from the outside in a similar manner to the structure of the seminiferous tubules in vivo. These structures of the organoids were developed at both temperatures (35 °C and 37 °C). Our results corroborate a previous study that found that the structure of the organoids was not affected by temperature (34 °C and 37 °C) [19]. Furthermore, our results show that some of the developed organoids in our system had a seminiferous tubule-like structure, since they contained peritubular cells outside the organoid and Sertoli cells and lumen inside. In addition, we demonstrated the presence of Leydig cells outside the organoids. These structures and cell composition may support the development of cells from the different stages of spermatogenesis in the organoids. Following the enzymatic dissociation of these organoids, we showed that they were composed of pre-meiotic (GFR-α, VASA) and meiotic (BOULE) cells in addition to CREM and post-meiotic (ACROSIN) cells. Thus, our results showed that some of the developed organoids are composed of a structure similar to that of the seminiferous tubule present in the testis of adult mice (the histology and type of cells (both somatic and germ cells) that composed these organoids). This structure may support in vitro the development of cells of the different stages of spermatogenesis. On the other hand, in our previous study, we could not clearly show the presence of lumen and Leydig cells in the developed organoids, but we showed, similar to this study, the presence of Sertoli cells, peritubular cells, meiotic and post-meiotic cells in some of the developed organoids [33]. Furthermore, our study compared, for the first time, the effect of temperature on the development of different stages of spermatogenesis, viability/apoptosis of the developed cells, and the functionality of Sertoli cells under a 3D in vitro system using different methods.

The development of cells in the different stages of spermatogenesis is also affected by incubation temperature. We showed that percentages and expression levels of the developed cells of the different stages of spermatogenesis were significantly higher at 35 °C than at 37 °C, especially after four weeks of culture. These results are consistent with those of a previous study which showed a significant decrease in the number of SYCP3 cells in the organotypic rat testicular organoid [19]. This could be caused by the damage of the first meiotic prophase, which increases DSBs and compromises DSB repair as demonstrated previously [24].

Our results are contradicting the findings of a previous study that showed a substantial increase in the expression levels of genes of undifferentiating spermatogonia such as Id4 and Thy-1, but a significant rise in the expression levels of differentiation-related genes such as c-kit, stra8, rec8, sycp3 and Ovll following the culture of SSCs at high temperatures (37 °C and 43 °C) [20]. This contradiction could be related to the different systems used. In their system, the authors of the aforementioned work used the CD1 SSC cell line and cultured them for a short time of 6 days in a two-dimensional culture, and they examined different markers from ours.

Our flow cytometry analysis confirmed the development of haploid cells (1N cells) and our immunostaining (CREM and ACROSIN-positive cells). However, our flow cytometry analysis showed no difference in the percentage of haploid cells at different temperatures. This could be related to the examination of all types of cells with 1N by flow cytometry and to technical elements that may lead to the loss of all types of cells (it is more potent in 1N cells, which are present in low amounts compared to other cells). We plan to examine these cells by FISH to validate our flow cytometry results. To further analyze the effect of temperature on developed round spermatids in vitro, we examined their effect on the expression levels of specific markers of round spermatids (ACRV1, SUN3) [34,35,36,37]. Our results suggested that under in vitro conditions, like in in vivo conditions, the lower temperature (35 °C) is better than the 37 °C one for the development of round spermatids, which may affect/regulate the development of complete spermatogenesis.

Sertoli cells are crucial testicular somatic cells for the development of complete spermatogenesis [3,38,39]. We suggested that temperature may affect Sertoli cell functionality. Our results showed that the expression levels of functional factors of Sertoli cell origin such as ABP, inhibin, AR, and GDNF that are involved in the development of spermatogenesis were significantly higher at 35 °C than at 37 °C. These results support the findings of previous studies showing that an increase in temperature affected Sertoli cell functionality, producing less ABP [25,40,41]. Furthermore, acute heat stress (37 °C, 0.5 h) decreased the viability of porcine immature Sertoli cells in vitro [42]. In addition, a single heat shock leads to changes in testicular weight and gene expression [43]. Our results suggest that high temperature may directly affect spermatogonial cells at different stages of differentiation, as well as impact the functionality of the supporting cells.

In summary, our results suggested that we can improve the conditions of developing different stages of spermatogenesis in vitro by incubating the cultures under 35 °C rather than 37 °C, mimicking in vivo conditions. These findings may deepen our understanding of the underlying mechanisms governing optimal conditions of normal spermatogenesis in vivo and in vitro and may inform the optimization of in vitro conditions of spermatogenesis aimed at preserving male fertility strategies for the future, particularly in prepubertal boys.

## 4. Materials and Methods

### 4.1. Animals

This study was performed in accordance with the Guiding Principles for the Care and Use of Research Animals Promulgated by the Society for the Study of Reproduction and was confirmed by the Ben-Gurion University Ethics Committee for Animal Use in Research (No. IL-16-04-2018). ICR male mice at different ages were used in the test (seven-day-old mice were employed to isolate cells from the seminiferous tubules for in vitro culture; 10 mice were used for each repeat), while seven-week-old mice were used as a positive control for immunofluorescence staining, tubule structure (Figure 2E,F), and flow cytometry analysis. The animals were purchased from Harlan Laboratories Israel Ltd., Jerusalem, Israel. They were housed in our animal house under normal conditions before being used according to the guidelines of the Ethics Committee for Animal Use in Research. A female was housed in one cage with litter. Cages were made of polysulfone, with a filter microisolator. Bedding consisted of sterile wood shavings. Enrichment comprised tissue paper, a paper bag and paper cup. The temperature range was around 20–26 °C. Humidity levels were around 30–70%. A regular light/dark cycle of 12 h each was applied. Diet comprised sterile SSNIFF, Germany. We used water bottles with sipper tubes.

### 4.2. Isolation of Seminiferous Tubular Cells

Testes from seven-day-old and seven-week-old mice were surgically removed. Thereafter, mechanical followed by enzymatic disintegration of the tissue was performed. The enzymatic digestion solution was composed of collagenase 1 (mg/mL) and DNAse I (1 mg/mL). Both collagenase and DNase were dissolved in PBS, and isolated cells were filtered through a sterile cell strainer (70 µM; BD Biosciences; San Jose, CA, USA) and centrifuged for 10 min at 1500 RPM (300× *g*) [27] After centrifugation, cells were diluted in 1 mL of medium (StemPro Thermo Fisher Scientific, Waltham, MA, USA, with 10% KSR Thermo Fisher Scientific Gibco, Waltham, MA, USA). The total number of viable cells, using trypan blue staining, was counted under a light microscope. For DNA content analyses, the cells were fixed in a 4% paraformaldehyde (PFA) solution (Santa Cruz, CA, USA) for 15 min in ice [33].

### 4.3. Methylcellulose Culture System (MCS)

Isolated cells from seminiferous tubules of immature mice were cultured in 24-well plates. Each well had 2 × 10^5^ cells/0.5 mL containing methylcellulose (R&D systems, Minneapolis, MN, USA), StemPro-34 medium, 10% KSR (Thermo Fisher Scientific Gibco, Waltham, MA, USA), and different growth factors such as human rEGF (recombinant epidermal growth factor) (20 ng/mL) (Biolegend, San Diego, CA, USA), human rGDNF (glial cell line derived nerve growth factor) (10 ng/mL) (Biolegend; San Diego, CA, USA), human rLIF (leukemia inhibitory factor) (10 ng/mL) (Biolegend; San Diego, CA, USA), and human rbFGF (basic fibroblast growth factor) (10 ng/mL) (Biolegend; San Diego, CA, USA). The cells were incubated for two to six weeks in a CO_2_ incubator at 37 °C and 35 °C.

### 4.4. Immunofluorescence Staining of the Cells

Slides of the cells that were isolated from the cultures after two to six weeks or from seminiferous tubules of immature mice (before culture; BC) were fixed with cold methanol. The cells were blocked with a blocking buffer consisting of 5% normal donkey serum (Jackson ImmunoResearch Laboratories, West Grove, PA, USA) for 30 min at room temperature. Thereafter, primary antibodies (Table 1) were added, and the slides were incubated overnight at 4 °C. The slides were washed with PBS, and the relevant fluorescent secondary antibodies (Table 1) were added to the cells on the slides and incubated for an additional 40 min at room temperature. Thereafter, the slides were washed with PBS, and DAPI (25 mg/mL) was added [33]. To calculate the percentage of positive-stained cells for each examined marker, we considered at least a total of 100 cells present in the examined field.

### 4.5. Confocal Microscopy

Organoids that were collected from the cultures were fixed in a 4% PFA solution and blocking was performed by the addition of a PBS-containing 5% donkey serum and a 0.5% triton and incubated for one hour at room temperature. Next, the relevant specific primary antibodies (Table 1) were added and incubated overnight at 37 °C. The slides were then washed three times in a PBS-containing 0.1% triton, followed by the addition of the relevant secondary antibodies (Table 1) and incubated for four hours at 4 °C. Thereafter, Hoechst-33342 (10 mg/mL) (Invitrogen, Waltham, MA, USA) was added for half an hour, followed by three PBS washes and the addition of a minimal solution of DAPI (25 mg/mL) (Santa Cruz Biotechnology, Santa Cruz, CA, USA). The stained organoids were analyzed using an FV1000 confocal microscope (Olympus, Tokyo, Japan) [43].

### 4.6. Acridine Orange (AO) and Ethidium Bromide (EtBr) Staining

Isolated cells from in vitro cultures in MCS were resuspended in 25 μL of PBS with 2 μL of a AO/EtBr solution (AO 100 μg/mL and EtBr 100 μg/mL in PBS). The cells were then visualized by fluorescence microscopy.

### 4.7. RNA Extraction and Real-Time PCR Analysis

For RNA extraction, we used the Dynabeads RNA direct kit (Dynal Biotech, Oslo, Norway) [33]. Real-time PCR quantitative analyses were performed using specific primers of different sequences (Table 2).

The reactions were introduced following the protocol of absolute qPCR SYBR Green mix (ABgene House, Blenheim Road, Epsom, UK) [33], including the amplification program and subsequent melting. GAPDH was used as a housekeeping gene. The result was analyzed using the 2^−ΔΔCt^ method.

### 4.8. DNA Content Analysis

Isolated organoids from the in vitro culture of MCS were enzymatically dissociated [33]. Thereafter, the cells were fixed in 4% paraformaldehyde (PFA) solution (Santa Cruz, CA, USA) for 15 min in ice, washed and then resuspended in PBS. Cells from immature (seven-day-old) mice and adult (seven-week-old) mice were used as controls. As a negative control for haploid cells, we used cells isolated from the seminiferous tubules of immature mice, and as positive control for haploid cells we used cells isolated from seminiferous tubules of adult mice. The protocol of DNA staining was performed [33]. The flow cytometry Canto II system (BD FACSAria III; software FACSDiva 8), a laser of 405 nm and a filter of 430–470 nm were used for the analysis. We used at least 5 × 10^5^ cells (the cells were adjusted after dissociation and filtered to avoid clumps). Data analysis was performed using flow v10.6.2 software.

### 4.9. Data Handling and Statistical Evaluation

All experiments were repeated three to four times. All data are expressed as the standard error of the mean (±SEM). Statistical significance analysis was performed with an unpaired Student’s *t*-test using Prism 9.3.1 (GraphPad Software, San Diego, CA, USA). *p* values below 0.05 were considered significant.

## 5. Conclusions

Our results suggest that we can improve the conditions for developing different stages of spermatogenesis in vitro by incubating the cultures under 35 °C rather than 37 °C, which mimics in vivo conditions. These conditions also affect the viability/apoptosis of the developed cells and the functionality of Sertoli cells. These findings may deepen our understanding of the mechanisms behind optimal conditions of normal spermatogenesis in vivo and in vitro and may assess optimization of the in vitro conditions of spermatogenesis for future male fertility preservation strategies, especially in prepubertal boys.

The limitations of the present study include the following aspects. (a) The use of a mouse system and extrapolating these findings to other species is questionable. (b) Some of the developed organoids exhibit similarities to seminiferous tubules; however, the in vitro conditions still do not fully replicate the complex microenvironment of the in vivo ones. (c) Duration of the cultures. (d) Translating findings into clinical applications requires additional research to establish the safety and efficacy of temperature modulation in human contexts.

## Figures and Tables

**Figure 1 ijms-25-02160-f001:**
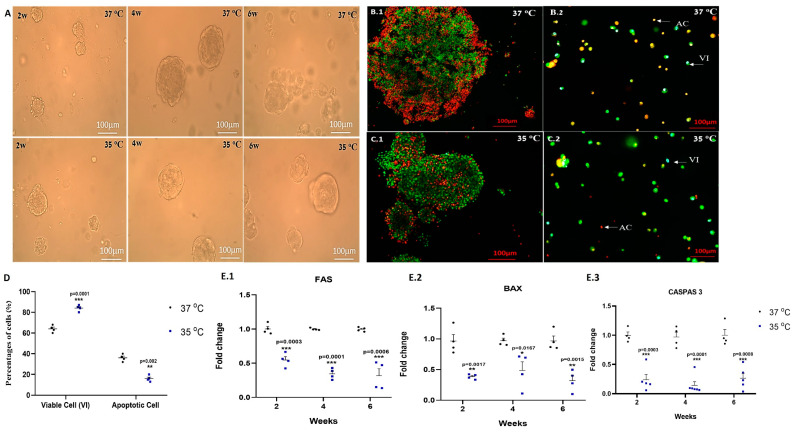
Effect of temperature on apoptosis of the growing cells in vitro. Isolated cells from seminiferous tubules of immature mice were cultured in methylcellulose-containing StemPro medium and growth factors (see the Section 4) at different temperatures (37 °C and 35 °C) for different lengths of time (two, four and six weeks). (**A**) Proliferation and differentiation of SSCs and development of organoids in MCS. (**B**,**C**) Organoids collected from cultures after four weeks at 37 °C and 35 °C, stained with propidium iodide and acridine orange before (**B.1**,**C.1**) and after enzymatic treatment (**B.2**,**C.2**). (**D**) The percentages of viable (VI; green color) and apoptotic cells (AC; orange and red color) that isolated after four weeks of culture and were evaluated after enzymatic treatment. (**E**) The developed cells and organoids were collected at different time points (two, four, and six weeks) to examine the gene expression of the apoptosis markers (FAS, BAX, CASPAS 3) (**E.1**, **E.2**, **E.3**, respectively) by qPCR analysis. The statistical significance comparison is between 37 °C and 35 °C at the same time point. * *p* < 0.05, ** *p* < 0.01, *** *p* < 0.001.

**Figure 2 ijms-25-02160-f002:**
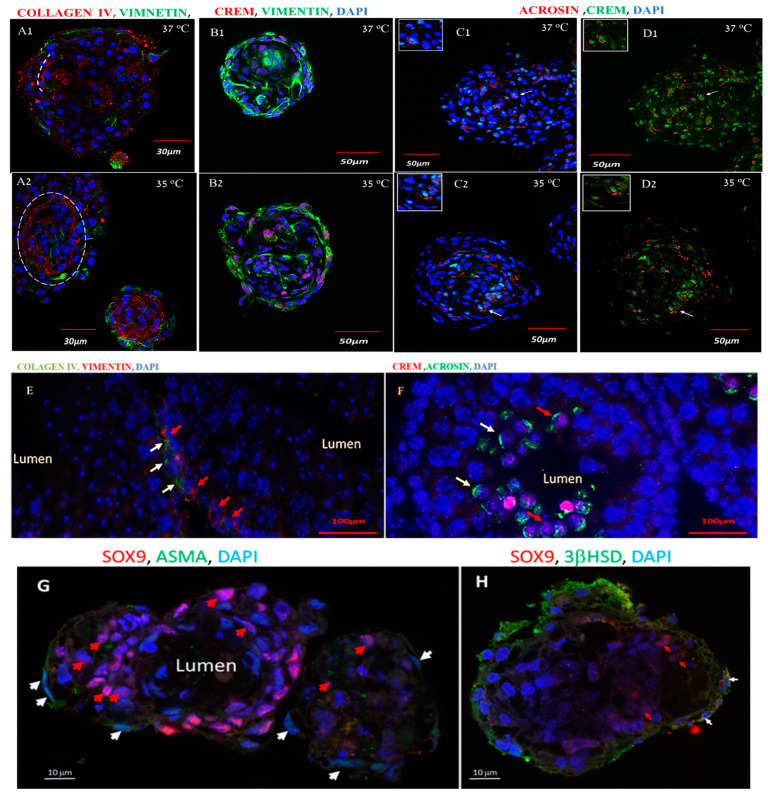
In Vitro development of organoids with a cellular composition similar to that pf seminiferous tubule structures. The developed organoids in MCS were collected after four weeks at 37 °C and 35 °C, fixed and stained using primary antibodies specific for the Sertoli cells (anti-vimentin antibodies; green color) and for the BM of the seminiferous tubules using collagen IV (red color) (**A1**, **A2**, respectively). The organoids were also examined for the presence of meiotic cells (CREM-positive cells; red color) (**B1**,**B2**) or for the presence of both meiotic cells (CREM-positive cells green color) and post-meiotic cells (ACROSIN-positive cells; red color) (**C1**,**C2**). Meiotic cells ((**D1**,**D2**) CREM-positive cells; green color) and post-meiotic cells (ACROSIN-positive cells; red color) without DAPI are displayed. As a positive control for the structure of the seminiferous tubules and stained cells, we used seminiferous tubules from adult mice. They were positively stained for collagen IV and vimentin ((**E**); red arrows indicate Sertoli cells and white arrows indicate collagen IV) and for ACROSIN and CREM ((**F**); red arrows indicate double-stained cells with ACROSIN and CREM, and white arrows indicate ACROSIN-stained cells). Organoid-like structures were positively stained for alpha-smooth muscle actin (ASMA) (green color; white arrows), a specific marker for peritubular cells (**G**); they also stained positively for SOX9 (red color, red arrows), a specific marker for Sertoli cells (**G**). A lumen (without staining for cells; no DAPI staining) was identified (lumen) (**G**). Outside the organoids, we identified Leydig cells, as they positively stained for 3βHSD (green color; white color arrows), a specific marker for Leydig cells (**H**). Inside this organoid, we identified Sertoli cells which positively stained for SOX9 ((**H**); red color arrows).

**Figure 3 ijms-25-02160-f003:**
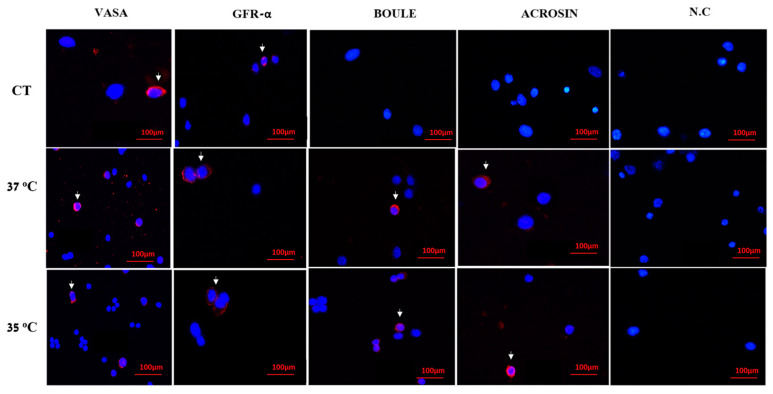
Effect of temperature on the development of different stages of spermatogenesis cells in vitro. Developed cells and organoids were collected BC in MCS after two to six weeks of culture at 37 °C and 35 °C and were fixed and stained with antibodies specific for cells of different stages of spermatogenesis: pre-meiotic markers (VASA and GFR-α), a meiotic marker (BOULE) and a post-meiotic marker (ACROSIN). Nucleus detection DAPI (blue) was used. Negative control (NC) was without the first antibody, BC—before culture. We mounted 100,000 cells/slide to quantify the results, as shown in Figure 4. White triangle marks cells stained positive for each marker.

**Figure 4 ijms-25-02160-f004:**
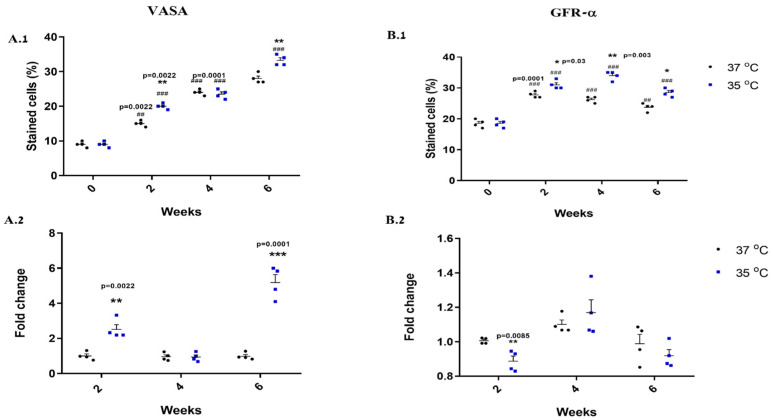
Effect of temperature on the development of pre-meiotic cells in vitro. Developed cells and organoids were collected after two, four and six weeks of culture at 35 °C and 37 °C to examine the presence and expression levels of VASA and GFR-α. This was achieved by IF staining and qPCR analyses using specific antibodies (**A.1**,**B.1**) and primers (**A.2**,**B.2**) for each cell/marker, respectively. To calculate the percentage of the positive-stained cells for each examined marker, we counted them from at least a total of 100 cells (presented in the examined field). Number of repeated experiments (N) = 4. Number of mice used in each repeated experiment (n) = 10. # Statistical significance between 35 °C and 37 °C at each time point was compared to that before culture (0)—## *p* < 0.01 and ### *p* < 0.001. * Statistical significance between 35 °C and 37 °C was compared at each time point—* *p* < 0.05 and ** *p* < 0.01 *** *p* < 0.001.

**Figure 5 ijms-25-02160-f005:**
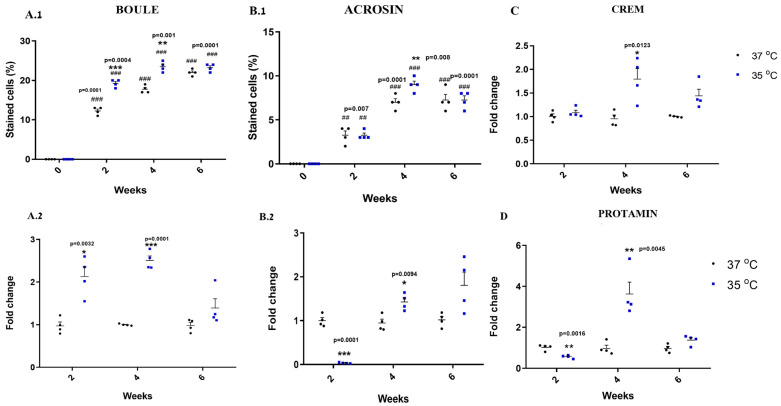
Effect of temperature on the development of meiotic/post-meiotic cells in vitro. Developed cells/colonies were collected after varying lengths of time (two, four, and six weeks) to examine the meiotic/post-meiotic stages of spermatogenesis using amino fluorescence staining with specific makers for BOULE and ACROSIN (**A.1**,**B.1**) and gene expression of BOULE, CREM, PROTAMINE, and ACROSIN (**A.2**,**B.2**,**C**,**D**) which was analyzed using qPCR. (0)–Before culture. Number of repeated experiments (N) = 4. Number of mice in each repeated experiment (n) = 10. #—Statistical significance between 35 °C and 37 °C at each time point was compared to that before culture (0). *—Statistical significance between 35 °C and 37 °C was compared at each time point. * *p* < 0.05, **## *p* < 0.01, ***### *p* < 0.001.

**Figure 6 ijms-25-02160-f006:**
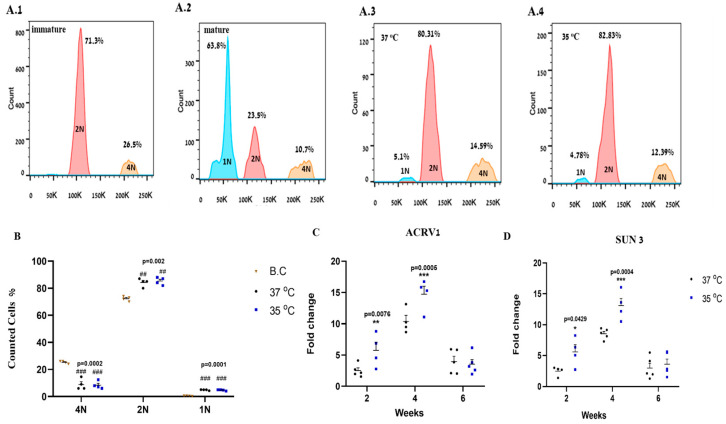
Effect of temperature on the development of round spermatid markers and on the DNA content of spermatogonial cells in vitro using flow cytometry analysis. Cells from developed organoids from both cultures that were incubated at 37 °C and 35 °C for 4 weeks were fixed and stained with Hoechst 33342 for DNA content analysis by flow cytometry (**A.3**,**A.4**). The cells were analyzed using the canto II flow cytometry machine. Cells isolated from the seminiferous tubules of immature (one-week-old) (**A.1**) and mature (seven-week-old) mice (**A.2**) were used to calibrate the system. 1N indicates haploid cells, 2N indicates diploid cells and 4N indicates tetraploid cells. (**B**) The quantification of repeats performed. (**C**,**D**) Developed cells/colonies were collected after different time points (two, four, and six weeks) and were examined for the presence of round spermatid markers ACRV1 and SUN 3 by analyzing their RNA expression. B.C—before culture. Number of repeated experiments (N) = 4. Number of mice in each repeated experiment (n) = 10. #—Statistical significance of DNA content was compared between 35 °C and 37 °C to BC (B). *—Statistical significance between 35 °C and 37 °C at each time point was compared for CRV1 (**C**) and SUN 3 (**D**). * *p* < 0.05, **## *p* < 0.01, ***### *p* < 0.001.

**Figure 7 ijms-25-02160-f007:**
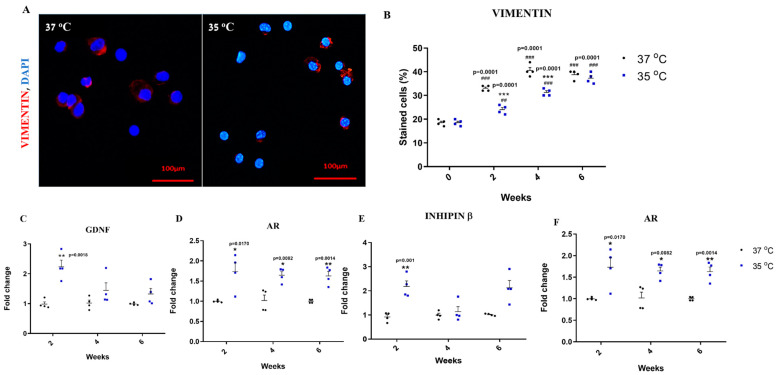
Effect of temperature on the development and functionality of Sertoli cells in vitro. Developed cells/colonies were collected after different time points (two, four and six weeks) to examine the presence of Sertoli cells using vimentin, the primary antibody specific for Sertoli cells (**A**,**B**). In addition, RNA was extracted to examine the expression levels of factors that indicate Sertoli cell activity (AR, ABP, INHIBIN and GDNF) by qPCR analysis using specific primers for each factor (**C**–**F**). #—Statistical significance between 35 °C and 37 °C at each time point was compared to that before culture (0), and *—Statistical significance between 35 °C and 37 °C was compared at each time point. ## *p* < 0.01 and ### *p* < 0.001. * *p* < 0.05, ** *p* < 0.01 and *** *p* < 0.001. Number of repeated experiments (N) = 4. Number of mice in each repeated experiment (n) = 8.

**Table 1 ijms-25-02160-t001:** Primary and secondary antibodies.

Antibody		Dilution	Catalog Number	Company
Collagen (rabbit)	Rabbit polyclonal anti-collagen IV	0.1111111	ab-6586	Abcam, Cambridge, UK
Vimentin (rabbit)	Rabbit polyclonal anti-vimentin	0.1111111	sc-7557	Santa Cruz Biotechnology, Santa Cruz, CA, USA
Vimentin (mouse)	Mouse monoclonal anti-vimentin	0.1805556	sc-6260	Santa Cruz Biotechnology, Santa Cruz, CA, USA
SOX9	Rabbit polyclonal anti-SOX9	1:50	ab185966	Abcam, Cambridge, UK
3βHSD	Goat polyclonal anti-3βHSD	1:50	sc30820	Santa Cruz Biotechnology, Santa Cruz, CA, USA
Alpha smooth muscle actin (ASMA)	Goat polyclonal anti-ASMA	1:50	ab21027	Abcam, Cambridge, UK
VASA	Rabbit Polyclonal anti VASA	0.1805556	NBP2-24558	NOVUS biologicals, Littleton, Centennial, CO, USA
GFR-α	Mouse monoclonal anti GFR-α	1:50	sc-271546	Santa Cruz Biotechnology, Santa Cruz, CA, USA
BOULE	Mouse monoclonal anti boule	0.1111111	sc-166660	Santa Cruz Biotechnology, Santa Cruz, CA, USA
CREM (rabbit)	Rabbit polyclonal anti CREM	0.1805556	12131-1-AP	ProteinTech Group, Chicago, IL, USA
CREM (mouse)	Mouse monoclonal anti CREM	1:50	sc-390426	Santa Cruz Biotechnology, Santa Cruz, CA, USA
ACROSIN	Rabbit polyclonal anti Acrosin	0.7361111	NBP-14260	NOVUS biologicals Littleton, Centennial, CO, USA
Cy3	Cy™3 AffiniPure F(ab’)₂ Fragment Donkey Anti-mouse\rabbit\goat IgG (H + L)	Mouse 1:1000	715-006-150	Jackson ImmunoResearch Laboratories, West Grove, PA, USA
		Rabbit 1:700	711-006-152	
		Goat 1:1000	705-546-147	
Alexa-flour 488	Alexa Fluor^®^ 488 AffiniPure F(ab’)₂ Fragment Donkey Anti-mouse\rabbit\goat IgG (H + L)	Mouse 1:100	715-006-150	
		Rabbit 1:200	711-006-152	
		Goat 1:200	705-546-147	

**Table 2 ijms-25-02160-t002:** Primers Used in Real-Time Quantitative PCR Analysis.

Primer	Gene Full Name	Primer Sequence
GAPDH	Glyceraldehyde 3-phosphate dehydrogenase	Fw-5′-ACCACAGTCCATGCCATCAC
		Rw-5′-CACCACCCTGTTGCTGTAGCC
VASA	ATP-dependent RNA helicase DDX4	Fw-5′-AGTATTCATGGTGATCGGGAGCAG
		Rw-5′-GCAACAAGAACTGGGCACTTTCCA
GFR-α	GFR-α	Fw-5′-CAGTTTTCGTCTGCTGAGGTTG
		RW-5-TTCTGCTCAAAGTGGCTCCAT
BOULE	boule homolog, RNA binding protein	Fw-5′-AACCCAACAAGTGGCCCAAGATAC
		Rw-5′-CTTTGGACACTCCAGCTCTGTCAT
CREM	cAMP responsive element modulator	Fw-5′-TTCTTTCACGAAGACCCTCA
		Rw-5′-TGTTAGGTGGTGTCCCTTCT
PROTAMINE	Protamine 1	Fw-5′-TCCATCAAAACTCCTGCGTGA
		Rw-5′-AGGTGGCATTGTTCCTTAGCA
ACROSIN	Acrosin prepropeptide	Fw-5′-TGTCCGTGGTTGCCAAGGATAACA
		Rw-5′-AATCCGGGTACCTGCTTGTGAGTT
ACRV1	Activin A receptor type I	FW-5′-GCTTCGGTTCAGCAACTTTC
		RW-5′-ACCACTCAGAGTCTTCTCATCTA
SUN-3	SUN domain-containing protein 3	FW-5′-GAAGCTGGGACCTCAGAAAG
		RW-5′-TATCCGGAGGCATCTCATAGT
AR	Androgen receptor	Fw-5′-TTGGGTGTGGAAGCATTGGA
		Rw-5′-TGGCGTAACCTCCCTTGAAA
Inhibin-β	Inhibin beta B chain	FW-5′-TCAGCTTTGCAGAGACAGAT
		RW-5′-TCTTGGAAGTACACCTTGACC
GDNF	Glial cell line-derived neurotrophic factor	FW-5′-GCCCCTGCTTTCTATCTGCT
		RW-5′-AGCCTTCTGAATGCGTGGTT
ABP	sex hormone binding globulin	Fw-5′-GCAGCATGAGGATTGCACTA
		Rw-5′-CATGAGGCTGGGGAATGTCT

## Data Availability

The data that support the findings of this study are available from the corresponding author upon reasonable request.

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
