# Peer review of "Effect of Temperature on the Development of Stages of Spermatogenesis and the Functionality of Sertoli Cells In Vitro"

_ijms, 2024, doi:10.3390/ijms25042160_

Round 1
Reviewer 1 Report (New Reviewer)
Comments and Suggestions for Authors
The conclusion should be more comprehensive.
Research limitations need to be explicitly stated.
The rationale behind selecting specific times and temperatures requires clarification.
How does the recent study differ in innovation from previous studies?
The objectives of the study should be more explicitly mentioned in the article's objectives
Comments on the Quality of English LanguageThe language of the article needs minor revisions
Author Response
The conclusion should be more comprehensive.
According to the valuable comments of the reviewer, we extended the conclusion section. See page #16, lines 497-500.
Research limitations need to be explicitly stated.
According to the valuable comments of the reviewer, we stated the possible research limitations. See page #16, lines 504-509.
The rationale behind selecting specific times and temperatures requires clarification.
According to the valuable comment of the reviewer, we added a stamen on pages #2-3, lines 98-101.
How does the recent study differ in innovation from previous studies?
According to the valuable comment of the reviewer, we added a statement on page #11, lines 342-348.
The objectives of the study should be more explicitly mentioned in the article's objectives
According to the valuable comment of the reviewer, we added a statement on page #1, lines 15-18.

Reviewer 2 Report (New Reviewer)
Comments and Suggestions for Authors
This manuscript describes an experiment for testing the in vitro effects of temperature in spermatogenesis and Sertoli cell functionality. The authors find that apoptosis and related markers increase with the higher temperature, mimicking in vivo effects.
This is an interesting study. However, revising previous studies (many cited by the authors), it seems that the use of a lower temperature has already been recommended for the in vitro culture of spermatogonia and Sertoli cells. The authors could explain in the introduction why they did not perform this kind of study previously (it seems that the culture system was established before 2012) and what is the novelty of that (if there is anything specific in their culture system, if the markers are new).
There are some other major issues critical regarding publication: Lack of detail in methods (experimental design, number of animals), statistics, figure quality.
There are some minor mistakes in the text. E.g., subscript for CO2 in L19 and elsewhere, spacing for °C (35 °C), superscript in L391 and elsewhere. The text could be improved by removing unnecessary helpers. For instance, "is known" in L53 ("is temperature-dependent...). Maybe in vivo/in vitro could be italicized?
More importantly, please revise the references; there are many mistakes.
There are also some minor language mistakes. Revise Methods, descriptions are sometimes convoluted or with minor grammar mistakes. Revise units (spacing, writing).
L84: "Recently," but the first reference is 12 years ago.
Methods: Please, revise for typographic errors, many affecting numbers and units.
In general, omit "as described previously by ... et al." and similar and just use the cite.
When referring to cell/object counts, indicate how many cells or objects were counted.
In 4.1, indicate clearly how many animals were allocated in the different experimental groups. Also, indicate if the animals were immediately used or if they were housed in your laboratory or animal house, conditions, cages, light/darkness times, feeding, animals/cage, etc.
L377: The use of the samples from animals of different ages is not clear. I understand that the younger mice were used for the culture and the older ones for obtaining seminiferous tubules, but this is not explicitly said. The authors must provide a good description of the experimental design clarifying this point. For instance, notice that you indicate "immature mice" in L401 half a page after first referring to these animals. Provide all the relevant information at once and in the proper places.
Moreover, there is no reference to the observation of cells from mature mice. There is a lack of detail on the methods and comparison between immature (cultured) and mature (directly observed, not used for culture?) testicular samples.
L379: Which DNAse and which concentrations?
L382: Should be rpm, but rather use xg.
L386: How were viable cells identified? Stained? Explain.
L395: Avoid breaking technical expressions (rbFGF).
L417-8: Concentrations?
L419: Mistake in model?
L440: Please, provide a full description of the flow cytometer, including lasers, detectors, cytograms, cells/min, and gating (whereas it might not be necessary here, please revise the MiFlowCyt guides 10.1002/cyto.a.20623). A screenshot of the panel could be added as supplementary material.
Statistics: Whereas a t-test could be good in some cases, an ANOVA is required when using different groups (you have a time x temperature model, not just two groups, as would be the case for a t-test). Please, revise for using proper statistical tests and posthoc comparison testing (for instance, you need a Dunnet's test in 2.4 for control-treatment contrasts), possibly with multiple comparison adjustments).
Results:
When showing specific P values above 0.001 (e.g., L106), please provide the exact P value (e.g., P=0.003), not P<0.05 or P<0.01 (this could be correct if referring to many variables at once, but, for individual comparisons, use the P value). Same comment for figures.
2.3: The authors indicate that they quantified the results (Fig. 3), but no plots or numbers are provided. This part must be revised and that information added.
Section 2.4 must be rewritten. It is very fragmentary, repetitive, and disorganized. Figure 4 is confusing and requires improvement both in the plots and caption (see below).
L208-9 (in figure caption): This is not clear, and it was not clearly explained in the methods.
L241: FACS is a trademark from BD. Use "flow cytometry" instead. Write Hoechst 33342 in full.
L243: Correct grammar mistakes (elsewhere).
L242-3: These details should be fully described in the experimental design in Methods.
For clarity, you could group by ploidy in Fig. 6B instead of grouping by temperature (you are comparing temperatures). Do the same in 6C-D, like in Fig. 5.
Figures:
The resolution is very poor (especially in Fig. 1, but also in the others), and its JPEG artifacts are evident. Images should be provided at a good enough resolution and in a lossless format (e.g., TIFF). The images are not acceptable as presented, but the editorial team could help you with that. I have some concerns with (possibly unintentional) alterations in the pictures: Labels in figures B1-C2 are sometimes vertically compressed, sometimes not in the same image! (C2) Please, revise and remake the composition. Take care to adjust the images to the same relative size. Crop the images for adjustment and, if necessary, use proportional scaling.
In any case, all original pictures should be available in supplementary material in case cropped/scaled images are presented in the manuscript. The authors should add the corresponding brightfield images too.
In general, labels and scales should be legible. Many are too small to see (and when zooming, a lack of resolution is evident).
Figures B1-C2 lack scale. Provide.
The columns in the plots are not necessary and obscure the results. Make the plots larger, the dots a bit smaller (so they do not overlap too much), and use only bars for SEM and a dot for the mean, removing the bars.
In plots, use "Fold change" instead of "Fold of increase." (?)
Change colors in Fig. 1D. Notice that red and green are a problem for a significant part of the population (revise guidelines for color blindness).
In the figures (photos and graphs), °C is presented with a superscript 0 (zero) and in some cases with a lowercase c. This is a minor issue, but for correctness, it should be a proper °C.
In the figures, please explain how the comparisons are carried out, not only the meaning of *. That is, "Asterisks indicate differences between temperatures within weeks: *p<0.05, 121 **p<0.01, ***p<0.001." Anyway, you can use P values in these cases (preferably on top and between the columns).
Figure description (captions) must be improved. In Figure 1D, properly describe the plot (when were these cells recovered, what relationship it has with the pictures. Describe abbreviations in all the captions (e.g., BC). The caption in Fig. 4 must be rewritten for clarity.
Figure 6A: The 3D plot obscures the position of the peaks while not contributing information. Use a 2D projection.
Fig. 6B: Do not break vertical axes (moreover, no need here).
Fig. 6C-D: I believe this is "fold change" in the y-axis like in previous figures? Be consistent with nomenclature (also, revise labels for clarity).
Fig. 7: Same comments.
Tables 1 and 2: If you include supplementary material with the original photos (strongly suggested), this table could be moved there. I also suggest removing horizontal lines and revising the formatting. In any case, reduce the font size and use the left justification in Table 2.
Discussion:
In general, the text would benefit from another language revision round. The discussion is mostly adequate, but by the second page, it loses detail and, in some parts, just repeats results with no effective discussion. Can be improved.
L299: Remove commas, modifying meaning.
L300: Not clear
L303: Please, refrain from odd and subjective expressions like "harmony."
L303 and following: I wonder if these results were already obtained in previous studies from the authors with the same model; and, if they differ, to provide some detail on that. That is, are the results novel beyond the temperature comparison?
L314-319: The same comment, was this already obtained in previous studies, or is this novel to the present study?
L322-324: Some citations supporting that? Is this novel for this study, or did the authors examine their model for these gene markers in previous experiments?
L325: Use the past tense.
L329: While it is adequate to refer to other studies as "of Sakib S et al.," etc., please keep that to a minimum and just use the sites.
L337: This deserves further clarification. Explain the differences in the systems and why the markers could differ. Indeed, the markers are different, and you did not study those. Explain why you chose a different set. Maybe you want to downplay the "contradictory" expression? I believe that unless you used the same (or, at least, some common) markers, you cannot claim that the results were opposite.
L338: Again, prefer "flow cytometry."
L341: Clarify "could be related to the examination of all types of cells with 1N by FACS." What alternative technique could be adequate for that? Moreover, develop the "technical elements that may lead to the loss of cells" (losing only N, or rather all kinds of cells?).
L342-345: Avoid repeating results verbatim.
L349-361: Again, repetition of introduction/results and very ineffective discussion. Avoid the passive with expressions such as "it was demonstrated." Notice that there are repetitions throughout the discussion in the form "it is possible that A occurs... thus, our results show that A occurs..."
L362: You only tested two temperatures, and with a limited set of markers; therefore, you cannot claim that you found "optimal" conditions, but that 35 °C might be more suitable than 37 °C. The remaining of the conclusion is good.
Please, refer to the general indications.
Author Response
This manuscript describes an experiment for testing the in vitro effects of temperature in spermatogenesis and Sertoli cell functionality. The authors find that apoptosis and related markers increase with the higher temperature, mimicking in vivo effects.
This is an interesting study. However, revising previous studies (many cited by the authors), it seems that the use of a lower temperature has already been recommended for the in vitro culture of spermatogonia and Sertoli cells. The authors could explain in the introduction why they did not perform this kind of study previously (it seems that the culture system was established before 2012) and what is the novelty of that (if there is anything specific in their culture system, if the markers are new).
According to the valuable comments of the reviewer, we added the information suggested. See page #2, lines 91-95
There are some other major issues critical regarding publication: Lack of detail in methods (experimental design, number of animals), statistics, figure quality.
According to the valuable comments of the reviewer, we added the information suggested in the relevant sections (see details below).
There are some minor mistakes in the text. E.g., subscript for CO2 in L19 and elsewhere, spacing for °C (35 °C), superscript in L391 and elsewhere. The text could be improved by removing unnecessary helpers. For instance, "is known" in L53 ("is temperature-dependent...). Maybe in vivo/in vitro could be italicized?
According to the valuable comments of the reviewer we added the information suggested in the relevant sections.
More importantly, please revise the references; there are many mistakes.
According to the valuable comment of the reviewer we corrected the references.
There are also some minor language mistakes. Revise Methods, descriptions are sometimes convoluted or with minor grammar mistakes. Revise units (spacing, writing).
According to the valuable comment of the reviewer we corrected the language.
L84: "Recently," but the first reference is 12 years ago.
Corrected, according to the valuable comment.
Methods: Please, revise for typographic errors, many affecting numbers and units.
Corrected, according to the valuable comment.
In general, omit "as described previously by ... et al." and similar and just use the cite.
Corrected, according to the valuable comment.
When referring to cell/object counts, indicate how many cells or objects were counted.
Corrected, according to the valuable comment. See page #7, lines 218-219, and page #13, lines 442-444.
In 4.1, indicate clearly how many animals were allocated in the different experimental groups.
Corrected, according to the valuable comment. See page #12, lines 399-401, in the relevant legend to figures.
Also, indicate if the animals were immediately used or if they were housed in your laboratory or animal house, conditions, cages, light/darkness times, feeding, animals/cage, etc
Corrected, according to the valuable comment. See page #12, lines 402-409.
L377: The use of the samples from animals of different ages is not clear. I understand that the younger mice were used for the culture and the older ones for obtaining seminiferous tubules, but this is not explicitly said. The authors must provide a good description of the experimental design clarifying this point. For instance, notice that you indicate "immature mice" in L401 half a page after first referring to these animals. Provide all the relevant information at once and in the proper places.
Corrected, according to the valuable comment. See page #12, lines 392-395.
Moreover, there is no reference to the observation of cells from mature mice. There is a lack of detail on the methods and comparison between immature (cultured) and mature (directly observed, not used for culture?) testicular samples.
Corrected, according to the valuable comment. See page #12, lines 401-402, and in relevant legend to figures.
L382: Should be rpm, but rather use xg.
Corrected, according to the valuable comment. See page #12, line 417.
L386: How were viable cells identified? Stained? Explain.
Corrected, according to the valuable comment. See page #12, lines 418-419.
L395: Avoid breaking technical expressions (rbFGF).
Corrected, according to the valuable comment. See page #13, line 430.
L417-8: Concentrations?
Corrected, according to the valuable comment. See page #13, lines 453-454.
L419: Mistake in model?
Corrected, according to the valuable comment. See page #13, line 455
L440: Please, provide a full description of the flow cytometer, including lasers, detectors, cytograms, cells/min, and gating (whereas it might not be necessary here, please revise the MiFlowCyt guides 10.1002/cyto.a.20623). A screenshot of the panel could be added as supplementary material.
Corrected, according to the valuable comment. See page #14, lines 482-484.
Statistics: Whereas a t-test could be good in some cases, an ANOVA is required when using different groups (you have a time x temperature model, not just two groups, as would be the case for a t-test). Please, revise for using proper statistical tests and posthoc comparison testing (for instance, you need a Dunnet's test in 2.4 for control-treatment contrasts), possibly with multiple comparison adjustments).
We thank the reviewer for the valuable comments. We used t-test since we always compared only two groups. In Fig. 1 we compared between 35 0C and 37 0C only in the same time point. In Fig. 4, we compare between 35 0C and 37 0C only in the same time point, or compared 35 0C in one time point with before culture or 37 0C in one time point with before culture. The same for other figures, as described in legend to each figure.
If the reviewer suggests to use additional statistical test, we will be happy to do that for validation of our results.
Results:
When showing specific P values above 0.001 (e.g., L106), please provide the exact P value (e.g., P=0.003), not P<0.05 or P<0.01 (this could be correct if referring to many variables at once, but, for individual comparisons, use the P value). Same comment for figures.
Corrected, according to the valuable comment. It is relevant only to Fig. 1D. The others have variables at once in the same figure.
2.3: The authors indicate that they quantified the results (Fig. 3), but no plots or numbers are provided. This part must be revised and that information added.
The quantified results and plots of figure 3 are presented in Fig.4.
Section 2.4 must be rewritten. It is very fragmentary, repetitive, and disorganized. Figure 4 is confusing and requires improvement both in the plots and caption (see below).
Corrected according to the reviewer comment.
L208-9 (in figure caption): This is not clear, and it was not clearly explained in the methods.
Corrected, according to the reviewer valuable comment. For each single experiment we used 10 immature mice. Each experiment was repeated 4 time to get validated results. See page # 7, lines 217-221.
L241: FACS is a trademark from BD. Use "flow cytometry" instead. Write Hoechst 33342 in full.
Corrected, according to the reviewer valuable comment. See page #9, lines 255-256.
L243: Correct grammar mistakes (elsewhere).
Corrected, according to the valuable comment.
L242-3: These details should be fully described in the experimental design in Methods.
Corrected, according to the reviewer valuable comment. See page #14, lines 479-484.
For clarity, you could group by ploidy in Fig. 6B instead of grouping by temperature (you are comparing temperatures). Do the same in 6C-D, like in Fig. 5.
Corrected, according to the reviewer valuable comment.
Figures:
The resolution is very poor (especially in Fig. 1, but also in the others), and its JPEG artifacts are evident. Images should be provided at a good enough resolution and in a lossless format (e.g., TIFF). The images are not acceptable as presented, but the editorial team could help you with that. I have some concerns with (possibly unintentional) alterations in the pictures: Labels in figures B1-C2 are sometimes vertically compressed, sometimes not in the same image! (C2) Please, revise and remake the composition. Take care to adjust the images to the same relative size. Crop the images for adjustment and, if necessary, use proportional scaling.
Thanks for the valuable comment. We did our best to provide figures with good resolution. The pictures B1-C2 present a merge (by the microscope) of colored cells, following acridine orange staining, that stained green, red, orange (the colors of live, dead and apoptotic cells, respectively). The stained cells are in suspension, and therefore some cells are at different levels (phase). The images are original with the relative size.
In any case, all original pictures should be available in supplementary material in case cropped/scaled images are presented in the manuscript. The authors should add the corresponding brightfield images too.
According to our response above, there is no need to add supplementary materials.
In general, labels and scales should be legible. Many are too small to see (and when zooming, a lack of resolution is evident).
Corrected, according to the reviewer valuable comment.
Figures B1-C2 lack scale. Provide.
Corrected, according to the reviewer valuable comment.
The columns in the plots are not necessary and obscure the results. Make the plots larger, the dots a bit smaller (so they do not overlap too much), and use only bars for SEM and a dot for the mean, removing the bars.
Partially corrected according to the reviewer valuable comments. It is difficult to remove the columns and show the results. The other comments were considered.
In plots, use "Fold change" instead of "Fold of increase." (?)
Corrected, according to the reviewer valuable comment. See all relevant figures.
Change colors in Fig. 1D. Notice that red and green are a problem for a significant part of the population (revise guidelines for color blindness).
Corrected, according to the reviewer valuable comment.
In the figures (photos and graphs), °C is presented with a superscript 0 (zero) and in some cases with a lowercase c. This is a minor issue, but for correctness, it should be a proper °C.
Corrected, according to the reviewer valuable comment. See all the relevant figures.
In the figures, please explain how the comparisons are carried out, not only the meaning of *. That is, "Asterisks indicate differences between temperatures within weeks: *p<0.05, 121 **p<0.01, ***p<0.001." Anyway, you can use P values in these cases (preferably on top and between the columns).
Corrected, according to the reviewer valuable comment. See all relevant figures. We explained the comparisons in the relevant legend to figures. The statistic comparisons between temperature are always for the same week and not between weeks.
Addition of the p values on top and between the columns will make the figures very crowded.
Figure description (captions) must be improved. In Figure 1D, properly describe the plot (when were these cells recovered, what relationship it has with the pictures. Describe abbreviations in all the captions (e.g., BC). The caption in Fig. 4 must be rewritten for clarity.
Corrected, according to the reviewer valuable comment.
Figure 6A: The 3D plot obscures the position of the peaks while not contributing information. Use a 2D projection.
Corrected, according to the reviewer valuable comment.
Fig. 6B: Do not break vertical axes (moreover, no need here).
Corrected, according to the reviewer valuable comment.
Fig. 6C-D: I believe this is "fold change" in the y-axis like in previous figures? Be consistent with nomenclature (also, revise labels for clarity).
Corrected, according to the reviewer valuable comment.
Fig. 7: Same comments.
Corrected, according to the reviewer valuable comment.
Tables 1 and 2: If you include supplementary material with the original photos (strongly suggested), this table could be moved there. I also suggest removing horizontal lines and revising the formatting. In any case, reduce the font size and use the left justification in Table 2.
Corrected, according to the reviewer valuable comment.
Discussion:
In general, the text would benefit from another language revision round. The discussion is mostly adequate, but by the second page, it loses detail and, in some parts, just repeats results with no effective discussion. Can be improved.
We considered the valuable comments of the reviewer and made changes accordingly.
L299: Remove commas, modifying meaning.
Corrected, according to the reviewer valuable comment. See page #10, line 315.
L300: Not clear.
Corrected, according to the reviewer valuable comment. See page #10, lines 314-316.
L303: Please, refrain from odd and subjective expressions like "harmony."
Corrected, according to the reviewer valuable comment. See page #11, line 319.
L303 and following: I wonder if these results were already obtained in previous studies from the authors with the same model; and, if they differ, to provide some detail on that. That is, are the results novel beyond the temperature comparison?
We thank the reviewer for the valuable comment. Our present results are published here for the first time. These results are novel in comparison the effect of temperature in our 3D in vitro culture system.
L314-319: The same comment, was this already obtained in previous studies, or is this novel to the present study?
Corrected, according to the reviewer valuable comment. See page #11, lines 342-348.
L322-324: Some citations supporting that? Is this novel for this study, or did the authors examine their model for these gene markers in previous experiments?
Corrected, according to the reviewer valuable comment. See page #11, lines 340-348.
L325: Use the past tense.
Corrected, according to the reviewer valuable comment. See page #11, line 338.
L329: While it is adequate to refer to other studies as "of Sakib S et al.," etc., please keep that to a minimum and just use the sites.
Corrected, according to the reviewer valuable comment. See page #11, lines 354, 356 and in other pages.
L337: This deserves further clarification. Explain the differences in the systems and why the markers could differ. Indeed, the markers are different, and you did not study those. Explain why you chose a different set. Maybe you want to downplay the "contradictory" expression? I believe that unless you used the same (or, at least, some common) markers, you cannot claim that the results were opposite.
Corrected, according to the reviewer valuable comment. See page #11, lines 357-363.
L338: Again, prefer "flow cytometry."
Corrected, according to the reviewer valuable comment. See page #11, lines 365, 367.
L341: Clarify "could be related to the examination of all types of cells with 1N by FACS." What alternative technique could be adequate for that? Moreover, develop the "technical elements that may lead to the loss of cells" (losing only N, or rather all kinds of cells?).
Corrected, according to the reviewer valuable comment. See page #11, lines 368-370.
L342-345: Avoid repeating results verbatim.
Corrected, according to the reviewer valuable comment. See pages #11-12, lines 372-375.
L349-361: Again, repetition of introduction/results and very ineffective discussion. Avoid the passive with expressions such as "it was demonstrated." Notice that there are repetitions throughout the discussion in the form "it is possible that A occurs... thus, our results show that A occurs..."
We considered the valuable comments of the reviewer and made changes accordingly. See page #12, lines 376-386.
L362: You only tested two temperatures, and with a limited set of markers; therefore, you cannot claim that you found "optimal" conditions, but that 35 °C might be more suitable than 37 °C. The remaining of the conclusion is good.
We changed the conclusion according to the reviewer suggestion. See page #12, lines 387-389.

Reviewer 3 Report (New Reviewer)
Comments and Suggestions for Authors
The research addresses the effect of temperature on the development of stages of spermatogenesis and the functionality of Sertoli cells in vitro. The study investigates how different temperatures impact the various stages of spermatogenesis and the functionality of Sertoli cells, providing insights into the potential implications for male fertility preservation strategies, especially in prepubertal boys. The topic is original and relevant at the same time. By focusing on the effect of temperature on the development of the stages of spermatogenesis and the functionality of Sertoli cells in vitro, the research addresses a specific gap in the field of reproductive biology and fertility preservation. This area of study is significant as it offers insights into the environmental factors that can affect male fertility and the potential implications for fertility preservation strategies, especially in prepubertal boys. Understanding the effects of temperature on spermatogenesis and Sertoli cell functionality is critical to developing optimal in vitro conditions for spermatogenesis and informing future male fertility preservation strategies. Therefore, this research fills a significant gap in the understanding of environmental factors affecting male fertility and the development of relevant preservation techniques. Previous studies have shown an association between elevated testicular temperatures and reduced sperm parameters such as motility and morphology, and this study further investigates the effects of temperature on the viability and functionality of Sertoli cells, which play a critical role in supporting spermatogenesis. The study examines the potential effects of temperature on male fertility preservation strategies, with a focus on prepubertal boys, an area that is relatively unexplored in the field of reproductive biology and fertility preservation. The language used is clear, objective and value-neutral, with a formal register and precise word choice. The structure is logical and follows a clear progression, with causal links between statements. The methodology used in the study appears to be well-designed and appropriate for the research question.
The conclusions drawn in the study are consistent with the evidence and arguments presented, and they effectively address the main question posed. These results support the conclusion that temperature has a significant impact on the development of stages of spermatogenesis and the functionality of Sertoli cells in vitro, which directly addresses the main question posed by the study. Therefore, the conclusions are well-supported by the evidence presented and effectively address the research question.
In addition to all these evaluations, the article is acceptable in its current form.
Author Response
The research addresses the effect of temperature on the development of stages of spermatogenesis and the functionality of Sertoli cells in vitro. The study investigates how different temperatures impact the various stages of spermatogenesis and the functionality of Sertoli cells, providing insights into the potential implications for male fertility preservation strategies, especially in prepubertal boys. The topic is original and relevant at the same time. By focusing on the effect of temperature on the development of the stages of spermatogenesis and the functionality of Sertoli cells in vitro, the research addresses a specific gap in the field of reproductive biology and fertility preservation. This area of study is significant as it offers insights into the environmental factors that can affect male fertility and the potential implications for fertility preservation strategies, especially in prepubertal boys. Understanding the effects of temperature on spermatogenesis and Sertoli cell functionality is critical to developing optimal in vitro conditions for spermatogenesis and informing future male fertility preservation strategies. Therefore, this research fills a significant gap in the understanding of environmental factors affecting male fertility and the development of relevant preservation techniques. Previous studies have shown an association between elevated testicular temperatures and reduced sperm parameters such as motility and morphology, and this study further investigates the effects of temperature on the viability and functionality of Sertoli cells, which play a critical role in supporting spermatogenesis. The study examines the potential effects of temperature on male fertility preservation strategies, with a focus on prepubertal boys, an area that is relatively unexplored in the field of reproductive biology and fertility preservation. The language used is clear, objective and value-neutral, with a formal register and precise word choice. The structure is logical and follows a clear progression, with causal links between statements. The methodology used in the study appears to be well-designed and appropriate for the research question.
The conclusions drawn in the study are consistent with the evidence and arguments presented, and they effectively address the main question posed. These results support the conclusion that temperature has a significant impact on the development of stages of spermatogenesis and the functionality of Sertoli cells in vitro, which directly addresses the main question posed by the study. Therefore, the conclusions are well-supported by the evidence presented and effectively address the research question.
In addition to all these evaluations, the article is acceptable in its current form.
We thank the reviewer for his positive comments.

Round 2
Reviewer 2 Report (New Reviewer)
Comments and Suggestions for Authors
The authors have considerably improved the manuscript. Although they have made a great effort with good improvements (especially in the discussion), some issues remain to be appropriately addressed.
There are some minor mistakes in the writing. Please check the spacing. Revise new additions (e.g., but elsewhere, L287-289).
For P-values, it is enough to write P<0.001 where appropriate.
Considering the counting, whereas it is a minor issue and acceptable for the current study, it is advisable to go for 200-400 counts for acceptable confidence. Notice that it would favorably reflect on the statistical analysis.
References: Still some minor mistakes; e.g., L597, italice; L551, L578, L604, capitals.
L202: Field > Sample (field is a single field of vision under the microscope, not the total sample).
L357: "Inconsistent" has a negative connotation. "Oppose" or "Contradict," maybe.
Again: In methods, omit "as described previously by ... et al." and similar and use the cite (a couple of occurrences).
L482-484: Improve, especially language. Specify gating and cell concentration (was it adjusted after dissociation?).
Specifically, did you check for doublets and cell aggregates? Please add an example flow cytometry panel in the supplementary materials showing all cytograms, histograms, and regions.
L484: What is the software name? Is it BD's?
Statistics: Although they could be improved, the explanation is acceptable. The exception is for the multiple comparisons against BC or 0 wk (e.g., Fig. 4-5, Fig. 6B, Fig. 7B). In this case, you are not comparing pairs but a treatment with many. As indicated in the first review, use Dunnet's. Please seek statistical advice if unsure.
Many figure issues have not been addressed (including supplementary material as requested). From my first report (adapted):
The resolution is very poor (especially in Fig. 1 and the others), and its JPEG artifacts are evident. Images should be provided at a good enough resolution and in a lossless format (e.g., TIFF). The images are not acceptable as presented, but the editorial team could help you with that.
In any case, all original pictures should be available in supplementary material, especially cropped/scaled images presented in the manuscript. The authors should also add the corresponding bright-field images.
In general, labels and scales should be legible. Many are too small to see (and when zooming, a lack of resolution is evident).
The columns in the plots are not necessary and obscure the results. Make the plots larger, the dots a bit smaller (so they do not overlap too much), and use only bars for SEM and a dot for the mean, removing the bars. In any case, the plots in Fig. 1 are very low quality and quickly lose quality if zoomed in. Plots in figures 4-6 are more acceptable, but in 7, they are still of insufficient quality. Please consult with the journal for resolution, size, and format.
In the figures (photos and graphs), °C is presented with a superscript 0 (zero) and, in some cases, with a lowercase c. This is a minor issue, but it should be properly written for correctness.
Some minor editing required.
Author Response
Comments and Suggestions for Authors
The authors have considerably improved the manuscript. Although they have made a great effort with good improvements (especially in the discussion), some issues remain to be appropriately addressed.
There are some minor mistakes in the writing. Please check the spacing. Revise new additions (e.g., but elsewhere, L287-289).
Corrected according to the reviewer's comment. Lines: #221, #250, #288,
For P-values, it is enough to write P<0.001 where appropriate.
Corrected according to the reviewer comment, in all relevant figures.
Considering the counting, whereas it is a minor issue and acceptable for the current study, it is advisable to go for 200-400 counts for acceptable confidence. Notice that it would favorably reflect on the statistical analysis.
We thank the reviewer for the comment and advice. We usually count more than 100 cells for validation of the results. See lines #202-203.
References: Still some minor mistakes; e.g., L597, italice; L551, L578, L604, capitals
Corrected according to the reviewer's comment. Lines: # 595-600, #605-608.
L202: Field > Sample (field is a single field of vision under the microscope, not the total sample).
Corrected according to the reviewer's comment. Lines # 202-203.
L357: "Inconsistent" has a negative connotation. "Oppose" or "Contradict," maybe.
Corrected according to the reviewer's comment. Line # 358.
Again: In methods, omit "as described previously by ... et al." and similar and use the cite (a couple of occurrences).
Corrected according to the reviewer's comment.
L482-484: Improve, especially language. Specify gating and cell concentration (was it adjusted after dissociation?)
Specifically, did you check for doublets and cell aggregates?
Corrected according to the reviewer's comment. Lines # 482-486.
L484: What is the software name? Is it BD's?
Please see lines 483-484.
Statistics: Although they could be improved, the explanation is acceptable. The exception is for the multiple comparisons against BC or 0 wk (e.g., Fig. 4-5, Fig. 6B, Fig. 7B). In this case, you are not comparing pairs but a treatment with many. As indicated in the first review, use Dunnet's. Please seek statistical advice if unsure.
We thank the reviewer for the valuable comment. In all our results and figures we always compared two parameters. We hope that our response is accepted.
Many figure issues have not been addressed (including supplementary material as requested). From my first report (adapted):
The resolution is very poor (especially in Fig. 1 and the others), and its JPEG artifacts are evident. Images should be provided at a good enough resolution and in a lossless format (e.g., TIFF). The images are not acceptable as presented, but the editorial team could help you with that.
We did our best to answer the reviewer's comment. We hope the resolution of the picture is OK now.
In any case, all original pictures should be available in supplementary material, especially cropped/scaled images presented in the manuscript. The authors should also add the corresponding bright-field images.
Original pictures were provided in supplementary material according to the reviewer's comment.
In general, labels and scales should be legible. Many are too small to see (and when zooming, a lack of resolution is evident.
Corrected according to the reviewer's comment.
The columns in the plots are not necessary and obscure the results. Make the plots larger, the dots a bit smaller (so they do not overlap too much), and use only bars for SEM and a dot for the mean, removing the bars. In any case, the plots in Fig. 1 are very low quality and quickly lose quality if zoomed in. Plots in figures 4-6 are more acceptable, but in 7, they are still of insufficient quality. Please consult with the journal for resolution, size, and format.
Corrected according to the reviewer's comment. We did our best to make the corrections, and hope it is now acceptable.
In the figures (photos and graphs), °C is presented with a superscript 0 (zero) and, in some cases, with a lowercase c. This is a minor issue, but it should be properly written for correctness.
Corrected according to the reviewer's comment.
This manuscript is a resubmission of an earlier submission. The following is a list of the peer review reports and author responses from that submission.
Round 1
Reviewer 1 Report
Comments and Suggestions for Authors
In their manuscript, the Authors evaluate the effect of body temperature on functional aspects associated with spermatogenesis. For this, the Authors use an in vitro model that, according to them, resembles the structure and function of seminiferous tubules. They have published the 'validation' of their in vitro model. However, this Reviewer believes that this model should be more characterized before is considered completely validated.
1. The Authors claim the presence of Sertoli cells in their model due to the expression of vimentin. However, contrary to what the Authors claim, this protein is not specific for Sertoli cells. Vimentin is a marker of somatic cells, and its expression does not indicate the unequivocal presence of Sertoli cells. The Authors must use specific markers for these cells, such as Sox9 or GATA4. Given the importance of Sertoli cells in the seminiferous epithelium, without ensuring the presence of these cells, the interpretation of the presented results is not valid.
2. Like what is commented above, the Authors must show that other cells that constitute the seminiferous tubules are present as well. Peritubular myoid cells are structural and functional component of the tubules. For instance, they not only form the wall of the tubules, but they also secrete factors that regulate spermatogenesis. The presence of these cells in the organoids must be determined.
3. The seminiferous tubules possess a well-defined microstructure of their epithelium. For instance, the presence of specific microenvironments (luminal and abluminal compartments) is essential for the function of the seminiferous epithelium. Why do the authors claim that their model resembles this epithelium without the presence of these compartments, structurally and functionally? Furthermore, the interstitial compartment is essential for the complete function of the seminiferous epithelium. In this, Leydig cells, and even macrophages, regulate spermatogenesis. Have the Authors considered the presence of these cells in their model? In other words, the model should resemble the testis microstructure, not only the one of seminiferous tubules.
In summary, without the complete validation of their model, the interpretation of the presented results is not, at this point, valid.
Author Response
Reviewer 1.
Comments and Suggestions for Authors
In their manuscript, the Authors evaluate the effect of body temperature on functional aspects associated with spermatogenesis. For this, the Authors use an in vitro model that, according to them, resembles the structure and function of seminiferous tubules. They have published the 'validation' of their in vitro model. However, this Reviewer believes that this model should be more characterized before is considered completely validated.
- The Authors claim the presence of Sertoli cells in their model due to the expression of vimentin. However, contrary to what the Authors claim, this protein is not specific for Sertoli cells. Vimentin is a marker of somatic cells, and its expression does not indicate the unequivocal presence of Sertoli cells. The Authors must use specific markers for these cells, such as Sox9 or GATA4. Given the importance of Sertoli cells in the seminiferous epithelium, without ensuring the presence of these cells, the interpretation of the presented results is not valid.
We thank the reviewer for the valuable comments. We added Fig. 2E which clearly shows the structure of seminiferous tubule with a positive staining for vimentin and collagen IV and shows the cellular composition similarity with our organoid-like structures. Furthermore, vimentin shows a specific staining for Sertoli cells in the seminiferous tubules.
- Like what is commented above, the Authors must show that other cells that constitute the seminiferous tubules are present as well. Peritubular myoid cells are structural and functional component of the tubules. For instance, they not only form the wall of the tubules, but they also secrete factors that regulate spermatogenesis. The presence of these cells in the organoids must be determined.
We thank the reviewer for the valuable comments. We agree with the reviewer for the important role that testicular somatic cells play in the process of spermatogenesis. We added Fig. 2E shows the structure of seminiferous tubule with a staining for vimentin and collagen IV showing cellular composition similarity with our organoid-like structures. In addition, in our previous publication (AbuMadighem, A., et al., 2022. Testis on a Chip - A Microfluidic 3-Dimensional Culture System for the Development of Spermatogenesis In-Vitro. Biofabrication. PMID: 35334473 DOI: 10.1088/1758-5090/ac6126) we showed the presence of aSMA (a specific marker for peritubular cells) in the developed organoids, and 3bHSD (a specific marker for Leydig cells) in cells isolated from organoids developed in our culture systems.
Due to the short time we had to respond to the reviewers’ comments we were unable to perform these experiments.
- The seminiferous tubules possess a well-defined microstructure of their epithelium. For instance, the presence of specific microenvironments (luminal and abluminal compartments) is essential for the function of the seminiferous epithelium.
Why do the authors claim that their model resembles this epithelium without the presence of these compartments, structurally and functionally?
We thank the reviewer for the valuable comments. We claim that using our system we were able to induce the development of stages of the spermatogenesis (we corrected the title of the paper according to the comments of the reviewer). We assume that we were not able to induce complete spermatogenesis because we did not provide all the optimal conditions needed for all the stages (including mature sperm) including the development of lumen.
Furthermore, the interstitial compartment is essential for the complete function of the seminiferous epithelium. In this, Leydig cells, and even macrophages, regulate spermatogenesis. Have the Authors considered the presence of these cells in their model? In other words, the model should resemble the testis microstructure, not only the one of seminiferous tubules.
Please see our response to the above comments (1-3).
In summary, without the complete validation of their model, the interpretation of the presented results is not, at this point, valid.

Reviewer 2 Report
Comments and Suggestions for Authors
Elevated temperature is known to negatively impact spermatogenesis in vivo. Whether it has a similar effect on spermatogonial stem cell differentiation and survival in 3D culture in vitro is the topic discussed in this manuscript. After isolating cells from the seminiferous tubules of immature mice and culturing them in methylcellulose, Jorban et al. measured cell apoptosis levels and characterized the developmental stages of meiotic germ cells using immunostaining and qPCR analysis. They concluded that cells growing at lower temperatures demonstrate better developmental potential.
There is not much novelty about this study, as the effect of higher temperature on spermatogenesis has been extensively studied both in vivo and in vitro. However, the results of this study may be worth reporting, given that it provides information on in vitro culturing of germ cells using the methylcellulose matrix. The current version of this manuscript requires extensive improvement on writing and data presentation. I have listed my comments below:
1. Improper and repetitive usage of words (detailed in language section)
2. Spermatogenesis refers to the process that germline stem cells develop to form sperm. Therefore, in the tile, “development of spermatogenesis” is not accurate.
3. Abstract should be a single paragraph summarizing what this study is about and what the conclusion is, not three separate paragraphs. Please shorten the second one (do not need to give a very detailed description about your findings here) and combine all three together.
4. Could you provide a higher resolution image for figure 1A? it’s very difficult to see the cells.
5. Figure 1B2 and C2, please provide images with cells zoomed in to allow better observation of the green and orange colored cells.
6. Figure 1E: please align the gene’s name “Fas, Bax and Caspase3” and make sure you use consistently either capitalized or non-capitalized, and also for the size of the font. This is also the case for figure 2, 4 and 5.
7. Figure 2: please provide an image showing immune stained seminiferous tubules to allow for the conclusion that the organoids developed in vitro showed similar structure.
8. Figure 2: please properly align A, B, Cand D in figure2 – note there’s no space between C and D. Also, please properly label D and describe it in the result section. It took me a while to figure out the relationship between 2C and 2D.
9. Figure 3: what are the red lines shown on each image? Is it a scale? If it is, please make sure they are consistent in their sizes and their position on the images.
10. Section 2.4, line 173- line 177, how many cells were quantified for the results shown in figure 4?
11. Figure 4B2, there’s obviously a reduction in GFR-alpha expression level for cells grown at 35C at 2 weeks as compared to those at 37C, based on the figure. However line 181-182 states “there’s no difference in the expression…” could you clarify and provide statistical analysis result?
12. Line 212: what about 6 weeks? It was not mentioned in the text.
13. Line 220: I don’t understand what is provided here on line 220.
14. Line 235: could you provide proper statistical analysis for the 2N cells that were grown at 35 and 37C ? it’s not clear to me that the increases were statistically significant based on graph 6B.
15. Where is figure 7?
16. There are two redundant references listed (13 and 14 are identical, 18 and 19 are identical)
Comments on the Quality of English Language
Writing requires significant revision. There are so many instances where the same phrase was repeated again and again. For example, you do not need to state 35C and 37C every single time in every single sentence. Also, you have listed the cellular marker for a particular type of meiotic cells repeatedly in a parathesis – this is not necessary. For example, because you have mentioned “Sertoli cells were stained with anti-vimentin in green” in line 130, there’s no need to repeat it again on line 133. This is just one example, but you have done this type of repetition everywhere throughout the manuscript. If you avoid repeating this information, this paper will read much smoother. Listed below are some other minor comments for grammar:
1. It’s not grammatically correct to say “development of spermatogenesis”. It’s redundant.
2. Line 14 and 16, there is no hyphen between in vivo and in vitro.
3. Line 21, “development”
4. Line 25, why use and/or? Didn’t you measure both?
5. Line 25-26, I don’t think you need to list the cellular markers for the cells at different developmental stages here.
6. Line 34, extra “.”
7. Line 36, change “of” to “for”
8. Line 45, change to niche for the completion of spermatogenesis.
9. Line 49-52, grammar for the sentence. Do you mean FSH affect different functions of Sertoli cells such as regulating the expression levels of … … …? ABP, AR… themselves are not a direct function of Sertoli cells.
10. Line 54-55, Spermatogenesis occurs….
11. Line 59: “increases” to “increased” or “elevated”
12. Line 67, need a space between “34C” and “compared”
13. Line 74, instead of “ram”, do you mean “men”?
14. Line 75-77, sentence grammar
15. Line 83-85, sentence grammar, and also you repeated “culture system 3 times in this sentence”!
16. Line 90, we evaluated the possible effect of temperature on the viability of cultured cells.
17. Line 97: remove “and incubation in” and add “at”
18. Line 98-99: our results showed that spheroids/organoids developed at different temperatures didn’t show differences in their size and shape.
19. Lien 101: remove “under conditions of 37C and 35C)
20. Line 103: please explain why need enzymatic treatment?
21. Line 106 – 108, redundant
22. Lien 121: “examine” instead of “examined”
23. Line 123: in virro development of organoids show a cellular composition similar to the seminiferous tubules.
24. Line 125: “culturing”
25. Line 128- 140: too repetitive in explaining what the cell markers are every single time.
26. Line 154 is not a complete sentence. You can say: we used anti-VASA and anti-GFR-alpha antibodies to identify cells at pre-meiotic stage. Same applies to the following sentence in line 155-156
27. Line 168: 2.4 title “ …….. pre-meiotic VASA and GFR-alpha positive cells in vitro”
28. Lien 170: at different temperatures, remove (35-37C), change to “significantly increased the unmber of pre-meiotic cells”
29. Line 173: GFR-alpha positive cells
30. Line 193-194, you do not need to list the cell markers in the parenthesis in the title
31. Line 196, add “cells” after post-meiotic. Remove (figure 5 A1 and B1 respectively)
32. Line 203, the percentage of Boule and Acrosin positive cells
33. Line 204, remove “figure 4A1 and B1)
34. Line 205, remove “(a meiotic stage cell marker)”
35. Lien 210, remove “(Aerosin and PROTAMIN)”
36. Line 230, change to “.” after (figure 6A1) and start a new sentence with “cells from seminiferous tubules”
37. Line 231, from the culture at different temperatures
38. Line 261, if this has any effect on the functionality of Sertoli cells present in in vitro cultures.
39. Line 281, and seminiferous tubule basement membrane, in which collagen type IV is one of the major components.
40. Line 285: at both temperatures.
41. Line 286: 37C
Author Response
Reviewer 2.
Comments and Suggestions for Authors
Elevated temperature is known to negatively impact spermatogenesis in vivo. Whether it has a similar effect on spermatogonial stem cell differentiation and survival in 3D culture in vitro is the topic discussed in this manuscript. After isolating cells from the seminiferous tubules of immature mice and culturing them in methylcellulose, Jorban et al. measured cell apoptosis levels and characterized the developmental stages of meiotic germ cells using immunostaining and qPCR analysis. They concluded that cells growing at lower temperatures demonstrate better developmental potential.
There is not much novelty about this study, as the effect of higher temperature on spermatogenesis has been extensively studied both in vivo and in vitro. However, the results of this study may be worth reporting, given that it provides information on in vitro culturing of germ cells using the methylcellulose matrix. The current version of this manuscript requires extensive improvement on writing and data presentation. I have listed my comments below:
We thank the reviewer for the valuable comments.
- Improper and repetitive usage of words (detailed in language section)
We made comprehensive English editing according to the reviewer comment.
- Spermatogenesis refers to the process that germline stem cells develop to form sperm. Therefore, in the tile, “development of spermatogenesis” is not accurate.
The title is corrected to “Effect of temperature on the development of stages of spermatogenesis and the functionality of Sertoli cells in vitro”.
- Abstract should be a single paragraph summarizing what this study is about and what the conclusion is, not three separate paragraphs. Please shorten the second one (do not need to give a very detailed description about your findings here) and combine all three together.
Corrected according to the reviewer comment.
- Could you provide a higher resolution image for figure 1A? it’s very difficult to see the cells.
Corrected according to the reviewer comment.
- Figure 1B2 and C2, please provide images with cells zoomed in to allow better observation of the green and orange colored cells.
Corrected according to the reviewer suggestion.
- Figure 1E: please align the gene’s name “Fas, Bax and Caspase3” and make sure you use consistently either capitalized or non-capitalized, and also for the size of the font. This is also the case for figure 2, 4 and 5.
Corrected according to the reviewer suggestion.
- Figure 2: please provide an image showing immune stained seminiferous tubules to allow for the conclusion that the organoids developed in vitro showed similar structure.
Images (E and F) provided according to the reviewer suggestion.
- Figure 2: please properly align A, B, Cand D in figure2 – note there’s no space between C and D. Also, please properly label D and describe it in the result section. It took me a while to figure out the relationship between 2C and 2D.
Corrected according to the reviewer comment.
- Figure 3: what are the red lines shown on each image? Is it a scale? If it is, please make sure they are consistent in their sizes and their position on the images.
Yes, these are scale bares. Corrected according to the reviewer comment.
- Section 2.4, line 173- line 177, how many cells were quantified for the results shown in figure 4?
Corrected according to the reviewer suggestion. The positive-stained cells for each examined marker were counted from at least a total of 100 cells.
- Figure 4B2, there’s obviously a reduction in GFR-alpha expression level for cells grown at 35C at 2 weeks as compared to those at 37C, based on the figure. However line 181-182 states “there’s no difference in the expression…” could you clarify and provide statistical analysis result?
Corrected according to the reviewer comment.
- Line 212: what about 6 weeks? It was not mentioned in the text.
Corrected according to the reviewer comment.
- Line 220: I don’t understand what is provided here on line 220.
Corrected according to the reviewer comment.
- Line 235: could you provide proper statistical analysis for the 2N cells that were grown at 35 and 37C ? it’s not clear to me that the increases were statistically significant based on graph 6B.
Corrected according to the reviewer comment.
- Where is figure 7?
Added.
- There are two redundant references listed (13 and 14 are identical, 18 and 19 are identical)
Corrected according to the reviewer comment.
Comments on the Quality of English Language
Writing requires significant revision. There are so many instances where the same phrase was repeated again and again. For example, you do not need to state 35C and 37C every single time in every single sentence. Also, you have listed the cellular marker for a particular type of meiotic cells repeatedly in a parathesis – this is not necessary. For example, because you have mentioned “Sertoli cells were stained with anti-vimentin in green” in line 130, there’s no need to repeat it again on line 133. This is just one example, but you have done this type of repetition everywhere throughout the manuscript. If you avoid repeating this information, this paper will read much smoother. Listed below are some other minor comments for grammar:
Corrected according to all the reviewer comments.
- It’s not grammatically correct to say “development of spermatogenesis”. It’s redundant.
- Line 14 and 16, there is no hyphen between in vivo and in vitro.
- Line 21, “development”
- Line 25, why use and/or? Didn’t you measure both?
- Line 25-26, I don’t think you need to list the cellular markers for the cells at different developmental stages here.
- Line 34, extra “.”
- Line 36, change “of” to “for”
- Line 45, change to niche for the completion of spermatogenesis.
- Line 49-52, grammar for the sentence. Do you mean FSH affect different functions of Sertoli cells such as regulating the expression levels of … … …? ABP, AR… themselves are not a direct function of Sertoli cells.
- Line 54-55, Spermatogenesis occurs….
- Line 59: “increases” to “increased” or “elevated”
- Line 67, need a space between “34C” and “compared”
- Line 74, instead of “ram”, do you mean “men”?
- Line 75-77, sentence grammar
- Line 83-85, sentence grammar, and also you repeated “culture system 3 times in this sentence”!
- Line 90, we evaluated the possible effect of temperature on the viability of cultured cells.
- Line 97: remove “and incubation in” and add “at”
- Line 98-99: our results showed that spheroids/organoids developed at different temperatures didn’t show differences in their size and shape.
- Lien 101: remove “under conditions of 37C and 35C)
- Line 103: please explain why need enzymatic treatment?
- Line 106 – 108, redundant
- Lien 121: “examine” instead of “examined”
- Line 123: in virro development of organoids show a cellular composition similar to the seminiferous tubules.
- Line 125: “culturing”
- Line 128- 140: too repetitive in explaining what the cell markers are every single time.
- Line 154 is not a complete sentence. You can say: we used anti-VASA and anti-GFR-alpha antibodies to identify cells at pre-meiotic stage. Same applies to the following sentence in line 155-156
- Line 168: 2.4 title “ …….. pre-meiotic VASA and GFR-alpha positive cells in vitro”
- Lien 170: at different temperatures, remove (35-37C), change to “significantly increased the unmber of pre-meiotic cells”
- Line 173: GFR-alpha positive cells
- Line 193-194, you do not need to list the cell markers in the parenthesis in the title
- Line 196, add “cells” after post-meiotic. Remove (figure 5 A1 and B1 respectively)
- Line 203, the percentage of Boule and Acrosin positive cells
- Line 204, remove “figure 4A1 and B1)
- Line 205, remove “(a meiotic stage cell marker)”
- Lien 210, remove “(Aerosin and PROTAMIN)”
- Line 230, change to “.” after (figure 6A1) and start a new sentence with “cells from seminiferous tubules”
- Line 231, from the culture at different temperatures
- Line 261, if this has any effect on the functionality of Sertoli cells present in in vitro cultures.
- Line 281, and seminiferous tubule basement membrane, in which collagen type IV is one of the major components.
- Line 285: at both temperatures.
- Line 286: 37C
All comment (1-41) were corrected according to the reviewer suggestions.

Round 2
Reviewer 2 Report
Comments and Suggestions for Authors
The authors have tried to address my comments. It would be helpful in the future to provide how you made changes directly under each comment than simply saying "corrected according to the reviewer comment"
Comments on the Quality of English LanguageLanguage still needs improvement.